# Healthy living environments create healthy physical activity-related habits (ELEVATE): A protocol for a smartphone-based mixed-method study

Marco Helbich[1]*, S.M. Labib[1], Isabel Liedtke[2], Danielle Sent[3], Ronald Crombag[4], Dick Ettema[1], Hanneke Posthumus[5,6], Saskia van der Zee[7], Mart Reiling[8], Susan Ruinaard[2], Anneke van Mispelaar[9], Aarnout Brombacher[10], Jeroen Hoyng[11], Bas Spierings[1], Wieger Savenije[2]

1 Human Geography and Spatial Planning, Utrecht University, Utrecht, The Netherlands, 2 Studio Bereikbaar, Rotterdam, The Netherlands, 3 Jheronimus Academy of Data Science, Eindhoven University of Technology, Tilburg University, 's-Hertogenbosch, The Netherlands, 4 De Boom en het Meer, Utrecht, The Netherlands, 5 Institute for Risk Assessment Sciences, Utrecht University, Utrecht, The Netherlands, 6 Data- en Kennishub Gezond Stedelijk Leven, Utrecht, The Netherlands, 7 Gemeentelijke of Gemeenschappelijke Gezondheidsdienst, Amsterdam, The Netherlands, 8 Track Landscapes, Utrecht, The Netherlands, 9 Bureau Buiten, Utrecht, The Netherlands, 10 Industrial Design, Eindhoven University of Technology, Eindhoven, The Netherlands, 11 Kenniscentrum Sport en Bewegen, Utrecht, The Netherlands

* m.helbich@uu.nl

## Abstract

Physical inactivity remains a key risk factor for adverse public health outcomes, disproportionately burdening lower socioeconomic groups. While individual-level interventions may temporarily increase walking-related physical activity (WPA) for some, they rarely achieve population-wide benefits. In contrast, built and natural environments are better positioned to support sustained WPA at the population level. Despite the importance of the environment to WPA, state-of-the-art research has limitations: it often relies on cross-sectional self-reports, uses coarse, simplistic environmental indicators, and focuses narrowly on residential neighborhoods, over-looking the diverse streetscapes people encounter in daily mobility. Using a transdisciplinary approach, ELEVATE addresses these gaps by 1) assessing how adults perceive and experience the built and natural streetscape environment during their WPA, 2) longitudinally assessing the relationships between streetscape environments and WPA, and 3) developing an artificial intelligence-powered system to suggest place-based environmental interventions that promote WPA. ELEVATE will employ a mixed-methods study design grounded in participatory co-creation, on-site and walk-along interviews to capture the everyday practices and experiences of WPA. We will use deep learning to extract objective environmental features and perceived streetscape qualities from street view images. A sample of approximately 200 adults residing in Amsterdam, The Netherlands, will be recruited. To examine relationships between micro-scale streetscape features and WPA, we will collect seven days of

**Data availability statement:** No datasets were generated or analysed during the current study. All relevant data from this study will be made available upon study completion.

**Funding:** MH led the consortium to obtain the funding. The proposal received funding from Health-Holland (HH-LoM-033). The funding organization did not play any role in the study design, data collection and analysis, decision to publish, or preparation of the manuscript.

**Competing interests:** The authors declare that some authors are employed by commercial organizations (Studio Bereikbaar, De Boom en het Meer, Track Landscapes, and Bureau Buiten). There are no patents, products in development, or marketed products associated with this research. This does not alter our adherence to PLOS ONE policies on data and materials sharing.

**Abbreviations:** WPA, walking-related physical activity; GPS, global positioning system; PA, physical activity; AI, artificial intelligence; GDPR, general data protection regulation; GIS, geographic information system; ST-DBSCAN, spatio-temporal density-based spatial clustering of applications with noise.

intensive longitudinal data using smartphones, including global positioning system (GPS)-based mobility and step counts. We will use spatiotemporal clustering to identify where people are most and least physically active, and mixed models to assess associations between these environmental measures and step counts. Qualitative and quantitative results will be triangulated in co-created focus groups. Finally, to support evidence-based public health and urban planning interventions, we will develop an AI-powered recommender system that generates context-sensitive (re) design guidelines framing the built and natural environment as a modifiable determinant of WPA. This study will generate evidence on how micro-scale built and natural street environments encountered during daily mobility are associated with WPA in adults. The findings are expected to inform place-based public health strategies by identifying modifiable environmental characteristics associated with physical activity and socioeconomic inequalities in urban settings.

## Introduction

### Background and rationale

Despite growing awareness, physical inactivity remains a major public health concern [1] and substantially contributes to the risk for premature mortality [2]. Many residents lack access to safe and pleasant public open spaces to be physically active, especially in lower-income neighborhoods [3]. Guidelines advise that adults should engage in at least 150 minutes of moderate-intensity physical activity (PA) per week [4]. However, adherence to these guidelines has shown little improvement. In the Netherlands, the proportion of adults meeting the PA recommendation in the post-COVID period remained between 43.1% in 2022 and 44.8% in 2024 [5]. If this modest upward trend continues, achieving the World Health Organization's global target of a 15% reduction in insufficient PA by 2030 will be challenging, particularly in the Dutch context [1].

Walking is a valuable source of daily energy expenditure and a key form of PA targeted in this study. To promote walking-related physical activity (WPA) more effectively and develop successful interventions, a robust understanding of the underlying determinants is required [6]. Although individual-level interventions (e.g., via smartphone apps and wearables) may boost short-term PA, they often fall short of driving population-level increases in WPA and disproportionately benefit already active or higher-socioeconomic status individuals, possibly exacerbating health inequalities [7].

Addressing upstream determinants at the neighbourhood level is more likely to achieve broad and lasting impacts on WPA [7,8]. Key upstream factors include the built and natural environment to which people are exposed, which can either facilitate or constrain opportunities for WPA behaviors [9,10]. As posited in the social ecological model of active living [11], reviews suggest that transport-related PA (e.g., walking) is often associated with the presence of retail services [12], higher residential density, pronounced mixed land-use, and better accessibility to everyday

destinations [13,14]. Recreational PA, by contrast, is inconsistently associated with the availability of natural environments (e.g., parks) and their aesthetic qualities, as well as sidewalks and street lighting [15]. Importantly, threshold effects may exist in these associations, where benefits are observed up to a certain level of exposure, beyond which effects may plateau or even reverse [16]. For example, an international study found that walking is positively associated with intersection density up to about 100 intersections per km$^2$, after which gains level off [16]. Recognizing these (non-linear) associations is crucial for informing context-sensitive policies and for providing evidence-based guidance to support municipalities in creating environments conducive to WPA.

## Knowledge gaps

Despite substantial research on the environmental determinants of WPA [10], several critical limitations persist. First, numerous studies rely on cross-sectional self-reports [17], which are subject to recall errors and social desirability bias, often leading to overestimation of PA levels [18]. Accelerometers, whether external devices or embedded in smartphones, provide more robust, highly accurate longitudinal measurements (e.g., step counts) in people's everyday settings. Relatedly, cross-sectional designs provide limited insight into the directionality of associations, making it difficult to determine whether the environment influences PA or whether observed relationships reflect residential self-selection (i.e., that highly active individuals self-select themselves into walkable neighborhoods), potentially inflating observed environmental effects [19].

Second, the built and natural environment is typically assessed within fixed geographic areas around participants' homes. A Dutch study found, however, that adults spend 42% of their time away from home, raising concerns about focusing solely on the residential context [20]. Thus, ignoring out-of-home environments encountered during daily activities leads to an incomplete understanding of which settings are most important for promoting PA. Given a 98% smartphone ownership in the Netherlands [21], global positioning system (GPS)-based tracking offers a reliable method to monitor mobility and WPA beyond the home.

Third, existing studies often rely on simplistic macro-level environmental proxies (e.g., population density) or walkability measures, usually assessed at the neighborhood scale [22], which fail to capture the micro-level streetscape qualities where most everyday environmental exposure occurs [23]. While environmental street audits provide micro-level data, their collection is time-consuming and resource-intensive, making them unsuitable for large-scale applications. Advances in artificial intelligence (AI) and the availability of street view imagery enable the automated, large-scale assessment of fine-grained streetscape features (e.g., pedestrian infrastructure, greenery, consumer services) [24], although these remain underutilized in WPA research [25].

Fourth, while quantitative studies on the association between environmental measures and WPA levels are highly valuable, relying solely on numerical data provides an incomplete picture. Quantitative methods can reveal patterns and associations, but they cannot fully capture how people perceive and experience their environment or how these perceptions and experiences shape daily routines and behaviors [26]. Qualitative insights are therefore essential to understand the lived experiences, meanings, and contextual factors that underlie the model estimates [27]. Combining both approaches in a mixed-methods design enables a more comprehensive, nuanced, and contextually grounded understanding.

Finally, a persistent gap remains between research evidence and its translation into urban design practice. Although studies suggest that street-level interventions (e.g., increased street trees) may promote WPA [28], how such evidence can be applied to redesign specific street contexts remains underdeveloped. Emerging AI-based approaches, including large language and world models, offer potential to support context-specific knowledge translation by linking environmental features to WPA [29,30]. However, such applications for urban street (re)design have not yet been developed or evaluated in the Dutch context.

## Objectives

To address these research gaps, we designed the 'Healthy Living Environments Create Healthy Physical Activity-related Habits' (ELEVATE) study, a smartphone-based mixed-methods study. This article presents the study protocol, summarizing the key aspects of how ELEVATE will be conducted between 2026 and 2030. Using mixed-methods within public-private partnerships, we aim to track adults' WPA longitudinally with smartphones, surveys, AI-based street view analyses, and on-site and walk-along interviews in Amsterdam, the Netherlands. Participatory co-creation with stakeholders will assist in grounding ELEVATE in lived experiences, thereby supporting the development of practical and context-sensitive solutions. ELEVATE's objectives are fivefold:

1) To understand how individuals perceive and experience the built and natural streetscape environment during their WPA;

2) To identify and co-create streetscape types and urban settings where adults are most and least engaged in WPA;

3) To assess how the built and natural streetscape environment encountered during daily mobility is (non-linearly) associated with WPA levels;

4) To examine whether environmental and WPA associations are moderated by behavioral and socio-demographic characteristics (e.g., income, education, gender);

5) To co-create urban streetscape design principles that stimulate the use of public spaces for WPA; and

6) To develop a recommender system targeted at urban (re)design suggestions to promote WPA.

## Materials and methods

ELEVATE combines community insights with advanced analytics to understand how (the design of) the built and natural environment influences WPA. Participatory and qualitative methods help identify relevant environmental factors, while geospatial and AI methods extract detailed features from street view images, and GPS, sensor, and survey data capture real-world activity patterns. These data streams are integrated to pinpoint which environmental elements support or hinder WPA. The findings are then translated into a context-sensitive urban (re)design recommender system, refined through ongoing user and stakeholder feedback. Through shared data and methods, collaboration with industry partners supports the development of more robust, policy-relevant models.

### Study site

Since approximately 90% of the Dutch population lives in urbanized areas [31], creating healthy living environments is a priority. Many environmental determinants of health are modifiable, offering a significant yet underutilized opportunity for preventive interventions to promote WPA. ELEVATE will focus on Amsterdam, where socio-economic and health inequalities are well-documented [32,33]. The city also offers rich diversity and exclusive street view data, with the municipality of Amsterdam strongly committed to healthy, data-driven planning. GPS-based studies routinely focus on a single city to capture high-resolution, context-rich data [28,34,35].

### Recruitment and survey design

Participants will be recruited through a multi-stage stratified sampling design. Alternatively, if recruitment proves challenging, as reported elsewhere [36,37], we will consider recruiting participants through an established panel. Eligibility criteria for study participation include: 1) age ≥18 years; 2) residence in a private household; 3) smartphone ownership; 4) adequate proficiency in the Dutch language; and 5) residence at the current address in Amsterdam for at least six months. To reach underserved groups (e.g., those with low socioeconomic status), we will collaborate with trusted neighborhood

ambassadors who will provide low-threshold and in-person support during enrollment. Ambassadors will be drawn from existing consortium partner networks and engaged purposely based on their local knowledge and community connections.

Potential participants will receive an information pack containing a cover letter, an information sheet detailing the study aim, and instructions on how to participate (e.g., how to conduct the survey, where to obtain the smartphone application, and where to carry the smartphone). Study materials will be in plain Dutch (and English) (B1-level) and reviewed for those with limited digital literacy. At the time of study enrollment, participants will also complete a baseline survey. The survey will gather detailed information on participants' demographics, health, residential preferences, etc. We will compare our sample with Statistics Netherlands data to ensure representativeness.

## Power analysis

We conducted a simulation-based power analysis to determine the sample size needed to assess associations between half-hourly exposures to built and natural environmental features (e.g., green space) and half-hourly physical activity levels (i.e., step counts). It was hypothesized, for example, that people exposed to more green spaces are more physically active [38]. We simulated data in which step counts are measured every half-hour during the day (12 hours) over seven consecutive days per participant (84 time points per person). The simulated outcome (i.e., steps per half-hour) was generated from a negative binomial distribution, and the predictor was, for example, green space exposure at the same half-hourly resolution.

Extracted from a meta-analysis pooling 42 studies [39], the baseline hourly step count was derived from a population mean of 9,448 steps/day (standard deviation = 2,218), distributed over 12 hours each day. Green space exposure was modeled as a continuous variable based on the normalized difference vegetation index. Based on a population-representative Dutch study [40], a mean vegetation index of 0.425 (standard deviation = 0.107) was assumed. We assumed a small effect size corresponding to log(1.15). We fitted a mixed negative binomial model with random participant-level intercepts. We used 500 simulations to estimate the power to detect, for example, the effect of green space.

With 180 participants, each providing 168 observations, the power to detect the assumed green space effect was estimated at 96.60% (confidence interval: 94.61%−98.01%). To accommodate potential participant drop-outs and missing data, we will target a minimum of 200 participants to ensure adequate power. Our targeted sample size is roughly comparable to other GPS-accelerometer studies [41,42]. The simulations were conducted in R 4.2 using the *simr* package [43].

## Co-creating physically active daily life environments

Co-creation with local stakeholders (e.g., planners, health professionals, municipal representatives, and residents) will be at the heart of ELEVATE, ensuring continuous, bidirectional learning between research and society. Its participatory approach fosters legitimacy, strengthens methods, and enhances ELEVATE's relevance. Embedding their perspectives will support inclusive knowledge production and scalable, just policy solutions for healthier cities. We will begin with a structured stakeholder mapping exercise to identify key actors across public, private, and civic domains.

We will organize 3–4 co-creation workshops [44]. These workshops, including focus groups, will identify the desirable attributes of WPA-supportive environments. Participants will also help with framing survey items. The resulting priorities will be translated into a ranked indicator list to guide the development of quantitative environmental indicators. Preliminary modeling results will be presented in interactive feedback sessions, where we can contextualize findings, identify blind spots, and suggest refinements. Similarly, user-centered testing of the AI-powered recommender system (in year 4) will involve design thinking sessions to assess usability, realism, and social fit of forward-looking intervention strategies.

### In-situ qualitative inquiries

Semi-structured on-site and walk-along interviews (i.e., a hybrid of in-depth interviews and participant observations) will be conducted [45,46]. These interviews are designed to gain a situated and detailed understanding of respondents' everyday interactions with and subjective perceptions and experiences of their local environments [47,48] with the interview data deeply informed by the environments in which they take place [46]. Specifically, we will discuss participants' daily activity environments and place meanings, daily physical activity routines, walking practices, and lived experiences, including perceived facilitators and barriers to walking in everyday life [49,50]. To document specific places and facilitate the interviews, photos will be taken [51]. After transcription and coding of the interviews, we will apply triangulation to compare findings and identify patterns of convergence and divergence in streetscape environments that support or hinder WPA.

The results will, amongst others, identify participant-defined environmental factors influencing walking practices and experiences. Qualitative findings on the lived experiences of streetscapes will also be mapped against quantitative project findings to contextualize and further explain the latter. A purposive sample, comprising approximately 25% of individuals from marginalized groups (e.g., those with low socioeconomic status or limited mobility), will be recruited through neighborhood ambassadors or trusted community intermediaries [52]. Based on a review [53], data saturation is expected to occur with approximately 20 participants.

### Micro-scale assessment of the built and natural environment

Based on the literature [23,54] and the co-creation sessions, we will compile multi-source, environmental indicators. For example, we will compute geographic information system (GIS)-based metrics for walking infrastructure, traffic speed, street connectivity, access to greenery, land-use mix, and the availability of public, social, and commercial amenities based on high-resolution remote sensing images (e.g., orthophoto), and other geographic information (e.g., OpenStreet-Map) [55]. While these measures provide a useful top-down view of the environment, we will complement them with people's street experiences at eye level. As successfully demonstrated in earlier research [56,57], we will use computer vision models to identify visible elements (e.g., trees) in publicly available street view images sourced from Mapillary [58]. After preprocessing (e.g., filtering unsuitable images), we will use deep learning models trained on annotated Cityscapes data to segment street scenes semantically.

Additionally, we will map human perceptions of streetscapes (e.g., safety, pleasantness, walkability) using state-of-the-art machine learning models trained on our over 20,000 unique crowdsourced stated preferences [59]. To validate the obtained ratings, domain experts will compare model outputs with manually annotated ground truth data. Given the critical role of street-level environments in encouraging WPA, these environmental indicators will be disaggregated at the street segment level rather than the neighborhood [60]. This micro-scale approach better captures movement impedance and more accurately reflects people's experiences navigating urban streets.

### Measuring PA under free-living conditions

Time-location-activity patterns are a key determinant of individuals' personal exposure [61]. Travel diaries are frequently used to collect WPA data [40]. Such reporting, typically conducted for one to three days, provides imprecise information on when and within which geographical context people are physically active, cannot replicate the route people have chosen, and the data quality appears to decline even after a single day [62]. In contrast, smartphones enable the continuous, passive collection of diverse behavioral, physiological, and environmental data that extend beyond the limits of traditional self-reports. Smartphones have become ubiquitous and are carried on the body throughout the day. The built-in sensors (e.g., accelerometer, GPS) have further enhanced the efficiency and feasibility of data acquisition, eliminating the need for user interaction and thereby minimizing the potential to alter human behavior.

We selected a smartphone app-based approach to collect high-quality, moment-by-moment WPA data unobtrusively, while minimizing participant burden and maintaining ecological validity [63,64]. Following recommendations [65], we will use step counts as a WPA measure. Step detection will use raw accelerometer signals, filtered through a low-pass filter to reduce noise. Validation studies show that smartphone accelerometry accurately measures WPA [66,67]. Mobility will be tracked via built-in GPS with location scans every, for example, 30 seconds [68]. GPS data will be cleaned by removing outliers (e.g., implausible speeds, horizontal errors >5 meters) and interpolated as needed to fill short gaps in coverage, according to best-practice guidelines [66], before transport modes are derived [20]. Raw accelerometer signals will be aggregated into short epochs along GPS-tracked paths [35], and step counts will be determined [69] (Fig 1). Moreover, the smartphone application facilitates the integration of experience sampling methods to capture individuals' real-time lived experiences [70].

Research suggests that a one-week period seems sufficient to capture routine activity patterns [64]. A push notification will signal study completion after seven consecutive days of data collection. The app will follow privacy-by-design principles. A pilot study involving volunteers (e.g., students) will assess the usability, battery life, and measurement accuracy of the app in laboratory-like settings. The resulting patterns will be benchmarked by the industry partners against established methods for analyzing active mobility and WPA.

We will assess smartphone wear-time compliance using established protocols [71]. Aligned with accelerometer studies, a valid smartphone wearing day has at least 8 hours [72]. Participants with fewer than four valid wearing days will be asked to proceed with the data collection to reach a full seven-day measurement period. If participant adherence to carrying the smartphone is insufficient at any point, or if the participant withdraws, a replacement participant will be recruited. Participants can pause or determine the tracking at any time. Accelerometer and GPS data will be matched and integrated with micro-scale built and natural environmental indicators using GIS. For recorded locations, exposures will be assigned based on street segment characteristics, with optional buffering (e.g., 25 m) to account for positional uncertainty. Environmental indicators will then be aggregated to the temporal resolution of the outcome (e.g., half-hour intervals) by computing time-weighted averages along individual mobility paths.

## Data analysis plan

Baseline descriptive statistics will be reported across the sample, including means, standard deviations, and frequencies. Baseline data will be statistically compared with $Chi^2$ and Wilcoxon tests across socioeconomic groups. Based on time-matched GPS-accelerometer data and high-resolution land-use data, we will map areas where active behaviors occur and identify locations where they should be encouraged. We will apply the spatio-temporal density-based spatial clustering

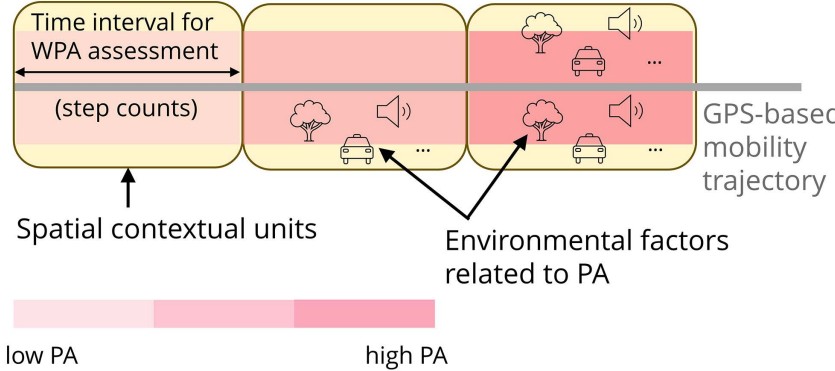

**Fig 1. Integration of GPS and step-count data.**

of applications with noise algorithm (ST-DBSCAN) [73] to identify frequently visited locations and commonly used movement paths [42]. This will distinguish between stationary clusters (e.g., home or work) and linear paths (e.g., commuting or recreational routes), providing profiles of where people are most and least physically active (e.g., underutilized streets and parks).

Using an intensive longitudinal design, we will assess 1) how characteristics of the built and natural environment are associated with half-hourly WPA levels (step counts) along people's daily mobility paths (Fig 1), and 2) how these associations vary by socioeconomic status. We will fit covariate-adjusted mixed negative binomial regression models as the primary inferential approach to account for repeated measurements of WPA within individuals [74]. Directed acyclic graphs, informed by theory and prior evidence, will be used to identify minimally sufficient sets of covariates (e.g., age, income) for regression adjustment [75,76]. To minimize biased estimates, we will control for individual preferences and attitudes (as reported in the survey) when selecting the home location [77]. Machine learning models (e.g., XGBoost) will be applied exploratorily to capture nonlinearity, identify variable importance, and model complex interactions among variables. Shapley values will ease model interpretation [78].

The on-site and walk-along interviews will be fully transcribed and analyzed using a thematic analysis approach. With the assistance of NVivo software, multiple rounds of deductive and inductive coding will be applied to identify overarching themes [79] – including recurring perceptions, experiences, and interactions with the features of the built and natural environment. A codebook will be developed to guarantee transparency, consistency and validity throughout the data analysis process. Resulting themes will be compared across participants and urban settings (e.g., neighborhoods, streetscapes) to highlight both shared patterns and place as well as participant-specific variations. The qualitative analysis will be set up in alignment with a reporting standard [80]. At the beginning of the project, on-site interviews will be organized to explore environmental factors influencing walking practices and experiences that need to be addressed with the baseline survey. Later in the project, walk-along interviews will be organized, and the results will be used to contextualize and 'spatially ground' statistical results as well as to further explain and validate quantitative findings.

## Recommender system for urban (re)design

We will develop a recommender system to generate evidence-based, equity-sensitive (re)design suggestions for WPA, particularly in low-activity and underserved neighborhoods. To ensure feasibility and reduce computational demands, we will build upon pre-trained/finetuned models, such as contrastive-language-image-pretraining [30], a learning transferable visual model [81], or a world model [29], using ELEVATE's multi-source quantitative and qualitative datasets. Training data will be balanced across neighborhood typologies to mitigate algorithmic bias. The system will generate context-specific (re)design suggestions by fusing spatial, morphological, visual, and textual data [82,83]. For example, it may be inferred that intersections without marked crossings are associated with low WPA, prompting targeted recommendations such as traffic-calming measures or improved wayfinding (Fig 2). We will focus on model validation and iterative refinement. Preliminary recommendations will be discussed in structured co-creation workshops that involve residents, urban planners, landscape architects, and other relevant stakeholders. Feedback will assess the feasibility, usability, and contextual relevance of the solution.

We will develop a publicly accessible, chat-based web interface (Fig 2). Users (e.g., local practitioners or residents) can upload (self-taken) street-level images or specify an address within the pilot area to receive interactive, explainable design recommendations. The backend will process these inputs and return reasoned interventions, grounded in empirical data and citizen input. The interface will support feedback loops for ongoing model refinement. The modeling pipeline will be hosted in an open-access repository (e.g., Hugging Face) with documentation to ensure scalability and reproducibility. The system will be piloted in Amsterdam, prioritizing neighborhoods where low WPA coincides with socioeconomic disadvantage, with potential roll-out to other cities to assess the model's generalizability and robustness across different urban morphologies.

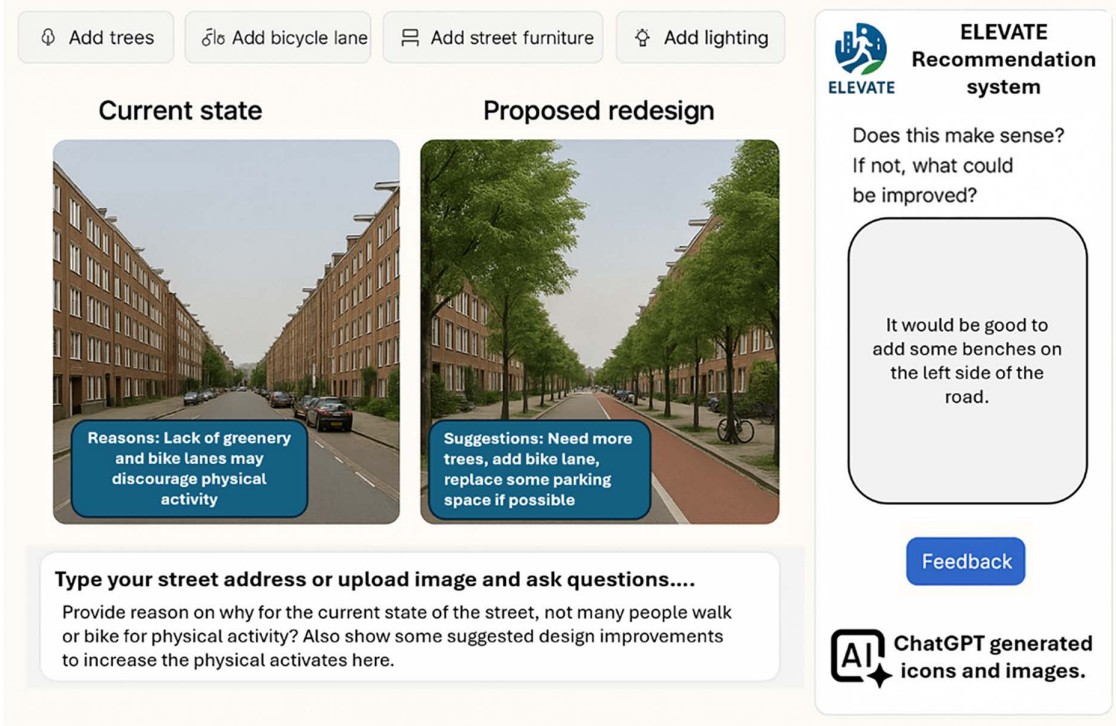

**Fig 2. A fictitious web interface of the (re)design recommender system.** The image is for illustrative purposes and was generated using ChatGPT 5.4 based on author-defined prompts and subsequent editing to reflect plausible real-world conditions. Fig 2 includes icons and hypothetical street-level images generated using ChatGPT 5.4. In accordance with the OpenAI Terms of Use (https://openai.com/policies/terms-of-use/), the authors retain ownership of the generated icons and images.

## Ethics

The legal basis for collecting and processing personal, privacy-sensitive data will be informed consent in accordance with the European General Data Protection Regulation (GDPR). Before participating in the study, each participant must provide informed consent. The consent procedure will ensure that all participants will be fully informed about the purpose of the research, the types of data collected, how their data will be processed, stored, and anonymized, and their rights under the GDPR, including the right to withdraw consent at any time without consequences. Any potential discomfort and risks associated with participation will be explained. Participants will provide consent via the smartphone application by selecting a checkbox; the application will automatically record the timestamp of consent. Separate consent will be obtained for participation in the focus group, for audio recording of the walk-along interview, and for the use of the transcript solely for analytical purposes. A separate consent form must be completed and signed for each component. To incentivize study participation and compensate them for their time and effort, shopping vouchers will be raffled.

For the processing of any sensitive data, the following measures will be applied to protect the participants' privacy: only data strictly necessary will be collected and processed, pseudonymization of personal data as soon as the data analyses strategy allows it, anonymization of survey responses and interview transcripts, GPS trajectories will not be shared publicly, separation of identifying information from research data, data encryption during transfer and storage, and access control restricted to authorized project team members only. The ethical and data protection aspects of ELEVATE were reviewed and approved by the Ethics Review Board of Utrecht University (Geo-25–0165).

## Data management

A data management plan has been developed and will be maintained and updated throughout the project. In compliance with Dutch and European regulations, the data security, storage, and protection strategies were reviewed and approved through a mandatory Privacy Impact Assessment conducted in consultation with the institutional Data Protection Officer at Utrecht University. The assessment was completed before data collection to ensure all privacy and data-handling risks were identified and mitigated. Briefly, GPS data collected via the smartphone app will be stored on GDPR-compliant servers hosted by SURF, UU, or TU/e. Data-sharing and processing agreements will be signed by project partners. All processing of personal data will be strictly in accordance with the GDPR and the privacy and information security policies of both universities. No personal identifiers (e.g., device IDs) other than those essential ones will be collected, and these will be pseudonymized immediately after collection. Interview transcripts will be anonymized and stored separately from the re-identification key. Survey responses will be stored separately from any technical identifiers and the GPS data. Access to direct participant identifiers and linkage keys will be restricted to authorized staff involved in the corresponding work package. Raw GPS data will not be published, as sensitive locations (e.g., home and work) could be inferred. To protect geoprivacy, data will be geomasked (e.g., by adding random noise) and/or aggregated to coarser spatial units prior to dissemination, preventing re-identification. Open data formats (e.g., from the Open Geospatial Consortium) will ease data exchange and platform-independent use. The data will be stored on secure servers, and data separation principles will be implemented to minimize the risk of re-identification. In line with the FAIR principles, environmental data will be made publicly available. Individual GPS tracks will never be shared publicly. Upon completion of ELEVATE, data will be retained for 10 years following the project's end to facilitate the replication and verification of the findings. All data will be transferred to Yoda, Utrecht University's infrastructure for integrated collaboration and long-term data storage.

## Discussion

### Practical and operational considerations

Implementing ELEVATE involves several practical considerations. First, recruiting and retaining a socioeconomically diverse sample is a key challenge in smartphone-based studies. To promote inclusiveness, recruitment will be supported by neighborhood ambassadors and trusted intermediaries, with study materials provided in plain Dutch (B1-level). Recruitment progress and attrition will be monitored, and strategies will be adjusted as needed to achieve the targeted sample size. Second, smartphone-based GPS and accelerometer data collection may be affected by battery consumption, operating system restrictions, signal loss, or inconsistent device carriage. To mitigate these issues, the data collection app will be piloted prior to the rollout, and participants will receive clear guidance on smartphone use. Data quality will be assessed using established wear-time and plausibility criteria, and participants with insufficient valid data may be invited to extend their participation or replaced. Third, integrating multiple data sources, including sensor data, surveys, environmental indicators, and qualitative data, will be addressed through standardized preprocessing pipelines and quality-control procedures. Finally, the AI-based recommender system will be developed using pre-trained models and iteratively refined through stakeholder and citizen feedback to ensure interpretability, feasibility, and relevance for public health practice.

### Dissemination strategies

ELEVATE's dissemination strategy is both multi-actor and multi-channel. Scientific dissemination will take place through the publication of peer-reviewed articles in open-access journals, supported by a commitment to open science and the FAIR principles. To engage professionals and decision-makers, co-creation workshops will be organized to share the findings. By collaborating closely with industry partners, we can move beyond traditional academic dissemination and engage policymakers, thereby translating emerging insights into action. For example, for industry and professional audiences,

demonstration sessions will showcase the recommender system, delivered as a user-friendly software-as-a-service platform. Public outreach activities will include roundtables, richly illustrated popular-science articles, infographics, and social media campaigns. A website will feature, for example, a story map to help residents explore relationships between local environments and WPA.

## Study limitations

While ELEVATE is among the first European mixed-method efforts to assess how objectively measured and subjectively experienced micro-environments shape daily WPA, several limitations should be acknowledged. First, smartphone-derived step counts may be less accurate than those from dedicated accelerometers, as they depend on participants' adherence to wearing protocols. Despite clear instructions, inconsistent device carriage and variability across devices and operating systems may result in misestimation of WPA. Besides careful validation against research-grade accelerometers, we may apply wear-time thresholds and restrict analyses to participants with high adherence to reduce this issue. Second, smartphone-based data collection may lead to selective participation and lower response rates, particularly among individuals with lower socioeconomic status and limited digital literacy, potentially affecting sample representativeness. To mitigate this risk, we will continuously cross-compare the sample characteristics with population data and, if needed, target specific groups. Third, as is typical for travel surveys, a one-week observation period may not capture seasonal or longer-term variability in WPA and environmental exposures. Fourth, Amsterdam's high urban density, mixed land use, and extensive active transport infrastructure may shape activity patterns, making the findings most applicable to similar urban contexts and warranting caution in lower-density or car-dependent settings. Although generalizability may be limited due to data collection in Amsterdam only, we will share tools and methods to facilitate replication in other settings. Fifths, short-term changes in the urban environment during the study period (e.g., construction, road closures, or new infrastructure) may influence WPA independently of the measured exposures. As these temporal dynamics are not captured in our geodata, this may introduce minor exposure misclassification. Sixth, the observational design precludes causal inference regarding the associations between micro-scale environments and WPA.

## Conclusions

The ELEVATE project will build a robust mixed-methods foundation to determine whether and to what extent micro-environmental conditions in the streetscape encountered during daily mobility influence WPA levels. By placing people's movements and lived experiences at the center of analysis, ELEVATE addresses critical gaps in current research on space-time built environmental exposures. By integrating smartphone sensing, GPS tracking, AI-powered streetscape analysis, a baseline survey, walk-along interviews, and co-creation workshops, the project will generate nuanced insights into how diverse urban settings shape WPA. These insights will inform evidence-based, equity-sensitive urban (re)design strategies that reflect the needs of marginalized communities and support more inclusive, walkable, and socially connected environments. The project will ultimately deliver actionable tools and guidance for researchers, policymakers, and planners who strive to create living environments that promote active living and advance healthier, more equitable urban futures.

## Acknowledgments

We gratefully acknowledge Paola Vasquez Ucho, Shristi Naresh Jain, and Vatsaid Molano Castro for their contributions to the development of the smartphone application architecture. We also appreciate the constructive comments provided by the proposal review panel and the two study protocol reviewers.

## Author contributions

**Conceptualization:** Marco Helbich, S. M. Labib, Isabel Liedtke, Danielle Sent, Ronald Crombag, Dick Ettema, Mart Reiling, Susan Ruinaard, Anneke van Mispelaar, Aarnout Brombacher, Bas Spierings, Wieger Savenije.

**Methodology:** Marco Helbich, S. M. Labib, Isabel Liedtke, Danielle Sent, Ronald Crombag, Dick Ettema, Hanneke Posthumus, Mart Reiling, Susan Ruinaard, Anneke van Mispelaar, Aarnout Brombacher, Bas Spierings, Wieger Savenije.

**Project administration:** Marco Helbich.

**Resources:** Marco Helbich.

**Visualization:** Marco Helbich, S. M. Labib.

**Writing – original draft:** Marco Helbich.

**Writing – review & editing:** Marco Helbich, S. M. Labib, Isabel Liedtke, Danielle Sent, Ronald Crombag, Dick Ettema, Hanneke Posthumus, Saskia van der Zee, Mart Reiling, Susan Ruinaard, Anneke van Mispelaar, Aarnout Brombacher, Jeroen Hoyng, Bas Spierings, Wieger Savenije.

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
