## [Decision Letter · Decision Letter 0]

26 Mar 2026

PONE-D-26-07970Healthy living environments create healthy physical activity-related habits (ELEVATE): A protocol for a smartphone-based mixed-method studyPLOS One

Dear Dr. Helbich,

Thank you for submitting your manuscript to PLOS ONE. After careful consideration, we feel that it has merit but does not fully meet PLOS ONE’s publication criteria as it currently stands. Therefore, we invite you to submit a revised version of the manuscript that addresses the points raised during the review process.

We look forward to receiving your revised manuscript.

Kind regards,

Dong Liu, Ph.D.

Academic Editor

PLOS One

Journal Requirements:

2. In the ethics statement in the Methods, you have specified that verbal consent was obtained. Please provide additional details regarding how this consent was documented and witnessed, and state whether this was approved by the IRB.

“The authors have declared that no competing interests exist.”

We note that one or more of the authors are employed by a commercial company: Studio Bereikbaar, De Boom en het Meer, Track Landscapes and Bureau Buiten.

6. We note that Figure 2 in your submission contain copyrighted images. All PLOS content is published under the Creative Commons Attribution License (CC BY 4.0), which means that the manuscript, images, and Supporting Information files will be freely available online, and any third party is permitted to access, download, copy, distribute, and use these materials in any way, even commercially, with proper attribution. For more information, see our copyright guidelines: http://journals.plos.org/plosone/s/licenses-and-copyright.

1. You may seek permission from the original copyright holder of Figure 2 to publish the content specifically under the CC BY 4.0 license.

Reviewers' comments:

Reviewer's Responses to Questions

Comments to the Author

1. Does the manuscript provide a valid rationale for the proposed study, with clearly identified and justified research questions?

Reviewer #1: Yes

Reviewer #2: Yes

2. Is the protocol technically sound and planned in a manner that will lead to a meaningful outcome and allow testing the stated hypotheses?

Reviewer #1: Yes

Reviewer #2: Yes

3. Is the methodology feasible and described in sufficient detail to allow the work to be replicable?

Reviewer #1: Yes

Reviewer #2: Yes

4. Have the authors described where all data underlying the findings will be made available when the study is complete?

Reviewer #1: Yes

Reviewer #2: Yes

5. Is the manuscript presented in an intelligible fashion and written in standard English?

Reviewer #1: Yes

Reviewer #2: Yes

6. Review Comments to the Author

You may also provide optional suggestions and comments to authors that they might find helpful in planning their study.

Reviewer #1: Thank you for inviting me to review this study protocol that seeks to an important public health and urban planning question: how micro-scale streetscape characteristics shape walking-related physical activity and how these relationships may differ across socioeconomic groups. The protocol is ambitious and has several notable strengths, including the integration of smartphone-based GPS and step-count data, detailed environmental exposure assessment using street-view imagery and GIS, and a mixed-methods design that combines quantitative analysis with qualitative interviews and co-creation elements. The emphasis on equity is also a clear strength. Overall, this is a strong and interesting protocol with a solid rationale, a well-designed mixed-methods approach, and clear relevance for both research and practice. Below please see my specific concerns.

1. In the data analysis plan section, the authors explain that GPS trajectories will be cleaned and matched with environmental indicators, which is helpful. It would further strengthen the protocol if the authors briefly clarify how exposures will be estimated. A short clarification here would improve transparency and reproducibility.

2. The manuscript already addresses ethics, pseudonymization, GDPR compliance, and secure storage, which is very good. Because GPS trajectories are particularly sensitive, it would be helpful to add one or two more sentences on practical geoprivacy protections, for example how home/work locations or highly identifying path information will be handled in derived datasets and outputs.

3. For the qualitative component, a brief note on coder procedures and how qualitative and quantitative findings will be integrated would make the protocol even clearer.

Reviewer #2: This research proposal employs a smartphone-based mixed-methods approach to investigate the association between urban streetscape environments and walking-related physical activity (WPA). The topic carries practical significance for both public health and urban planning, and puts forward systematic solutions to critical gaps in existing research regarding data collection, environmental assessment scales, study design, and research translation.

The following revision suggestions are provided for the authors’reference.

1. (Introduction) (Background and rationale): On page 3, it is mentioned that “threshold effects may exist in these associations”, yet no specific examples of environmental characteristic thresholds (e.g., thresholds for green space coverage or street connectivity) are cited from existing literature. It is recommended to supplement 1–2 typical cases to lay a more solid foundation for subsequent research hypotheses and analyses.

2. Objectives and hypotheses: Hypothesis 4 states that “co-created urban streetscape design principles will increase WPA usage in public spaces”, but does not specify the methods for testing this hypothesis (e.g., pre–post comparison, simulation application evaluation). It is recommended to add such details to align research hypotheses with subsequent methods.

3. (Study site): Only the research advantages of Amsterdam are described (clear socioeconomic and health inequalities, abundant streetscape data), without addressing the potential impacts of the city’s unique streetscape characteristics (e.g., high-density neighbourhoods, extensive bicycle lane networks) on WPA. It is recommended to supplement this background and clarify the generalisability boundaries of the study findings.

4. Recruitment and survey design: It is mentioned that “community ambassadors will be engaged to recruit vulnerable groups”, but the selection criteria for community ambassadors and recruitment incentives are not specified. It is recommended to supplement these details to ensure recruitment feasibility and sample diversity.

5. Data analysis plan: It is noted that “directed acyclic graphs (DAGs) will be used to identify potential covariates from questionnaires”, yet the core variable set and construction method for DAG development are not explained; this content should be added. Meanwhile, the analytical division of labour between mixed-effects negative binomial regression models and XGBoost models is not clarified (e.g., the former for testing linear associations and the latter for exploring nonlinear and interactive effects). It is recommended to refine the logic of model application.

6. Practical and operational considerations: It is mentioned that “smartphone data collection is affected by battery consumption and signal loss”, but no specific technical solutions are proposed (e.g., low-power collection modes, offline data storage), with only pilot testing mentioned. It is recommended to supplement targeted mitigation measures to enhance the study’s operability.

7. (Conclusions): The conclusion only emphasises the research value for “equitable, walkable urban environments” without explicitly acknowledging study limitations (e.g., the sample being limited to Amsterdam, impacts of urban environmental changes during the study period). It is recommended to add a limitation analysis to make the conclusions more objective.

7. PLOS authors have the option to publish the peer review history of their article (what does this mean?). If published, this will include your full peer review and any attached files.

Do you want your identity to be public for this peer review? For information about this choice, including consent withdrawal, please see our Privacy Policy.

Reviewer #1: No

Reviewer #2: No

---

## [Author Response · Author response to Decision Letter 1]

2 Apr 2026

Response (PONE-D-26-07970)

We thank the editor, Prof. Dong Liu, and the two reviewers for their insightful and constructive feedback, which has improved our manuscript. Our point-by-point responses to their comments are provided below.

Journal Requirements

Response: Done. The revised manuscript follows the style requirements.

Response: Done. The revised manuscript follows the style requirements.

2. In the ethics statement in the Methods, you have specified that verbal consent was obtained. Please provide additional details regarding how this consent was documented and witnessed, and state whether this was approved by the IRB.

Response: Done. The subjects agree verbally, but they still need to sign the consent form. The sentence was rephrased as follows: “Participants will provide consent via the smartphone application by selecting a checkbox; the application will automatically record the timestamp of consent. Separate consent will be obtained for participation in the focus group, for audio recording of the walk-along interview, and for the use of the transcript solely for analytical purposes. A separate consent form must be completed and signed for each component. […] The ethical and data protection aspects of ELEVATE were reviewed and approved by the Ethics Review Board of Utrecht University (Geo-25-0165).”

Response: Confirmed. We only mention the funding in the funding statement.

“The authors have declared that no competing interests exist.”

We note that one or more of the authors are employed by a commercial company: Studio Bereikbaar, De Boom en het Meer, Track Landscapes and Bureau Buiten.

Response: Done. Adjusted as follows: “The ELEVATE project is funded by Health-Holland, Top Sector Life Sciences & Health (grant number HH-LoM-033). The project underwent independent peer review as part of the funding process. All project partners, including academic and commercial organizations (Studio Bereikbaar, De Boom en het Meer, Track Landscapes, and Bureau Buiten), receive funding through this consortium. Health-Holland provides financial support in the form of authors’ salaries and/or research materials but does not have a direct role in the study design, data collection and analysis, decision to publish, or preparation of the manuscript. The commercial partners were involved in developing the research proposal and will contribute to the conduct of the study. The specific roles of all authors, including those affiliated with commercial organizations, are described in the Author Contributions section.” The author's contributions statement accurately reflects each person's contributions.

Response: Done. We updated the statement as follows: “The authors declare that some authors are employed by commercial organizations (Studio Bereikbaar, De Boom en het Meer, Track Landscapes, and Bureau Buiten). There are no patents, products in development, or marketed products associated with this research. This does not alter our adherence to PLOS ONE policies on data and materials sharing.”

Response: Corrected. We copied the ethics and data management section to the Methods section as requested.

6. We note that Figure 2 in your submission contain copyrighted images. All PLOS content is published under the Creative Commons Attribution License (CC BY 4.0), which means that the manuscript, images, and Supporting Information files will be freely available online, and any third party is permitted to access, download, copy, distribute, and use these materials in any way, even commercially, with proper attribution. For more information, see our copyright guidelines: http://journals.plos.org/plosone/s/licenses-and-copyright. We require you to either (1) present written permission from the copyright holder to publish these figures specifically under the CC BY 4.0 license, or (2) remove the figures from your submission:

1. You may seek permission from the original copyright holder of Figure 2 to publish the content specifically under the CC BY 4.0 license. We recommend that you contact the original copyright holder with the Content Permission Form (http://journals.plos.org/plosone/s/file?id=7c09/content-permission-form.pdf) and the following text:

Please upload the completed Content Permission Form or other proof of granted permissions as an "Other" file with your submission. In the figure caption of the copyrighted figure, please include the following text: “Reprinted from [ref] under a CC BY license, with permission from [name of publisher], original copyright [original copyright year].”

Response: Added. The figure images are not sourced from copyrighted material. They illustrate a conceptual AI-driven recommendation system; no such system has been developed to date. The images were generated using ChatGPT-4.5 based on author-defined prompts and subsequent editing to reflect plausible real-world conditions. The authors take full responsibility for these generated images, which are shared under the CC BY 4.0 license. We added the following to the figure caption: “The image is for illustrative purposes and was generated using ChatGPT-4.5 based on author-defined prompts and subsequent editing to reflect plausible real-world conditions.”

Response: Done. Thanks for pointing this out.

Reviewers' comments

1. Does the manuscript provide a valid rationale for the proposed study, with clearly identified and justified research questions? The research question outlined is expected to address a valid academic problem or topic and contribute to the base of knowledge in the field.

Reviewer #1: Yes Reviewer #2: Yes

2. Is the protocol technically sound and planned in a manner that will lead to a meaningful outcome and allow testing the stated hypotheses? The manuscript should describe the methods in sufficient detail to prevent undisclosed flexibility in the experimental procedure or analysis pipeline, including sufficient outcome-neutral conditions (e.g. necessary controls, absence of floor or ceiling effects) to test the proposed hypotheses and a statistical power analysis where applicable. As there may be aspects of the methodology and analysis which can only be refined once the work is undertaken, authors should outline potential assumptions and explicitly describe what aspects of the proposed analyses, if any, are exploratory.

Reviewer #1: Yes Reviewer #2: Yes

3. Is the methodology feasible and described in sufficient detail to allow the work to be replicable? Descriptions of methods and materials in the protocol should be reported in sufficient detail for another researcher to reproduce all experiments and analyses. The protocol should describe the appropriate controls, sample size calculations, and replication needed to ensure that the data are robust and reproducible.

Reviewer #1: Yes Reviewer #2: Yes

4. Have the authors described where all data underlying the findings will be made available when the study is complete? The PLOS Data policy requires authors to make all data underlying the findings described in their manuscript fully available without restriction, with rare exception, at the time of publication. The data should be provided as part of the manuscript or its supporting information, or deposited to a public repository. For example, in addition to summary statistics, the data points behind means, medians and variance measures should be available. If there are restrictions on publicly sharing data—e.g. participant privacy or use of data from a third party—those must be specified.

Reviewer #1: Yes Reviewer #2: Yes

5. Is the manuscript presented in an intelligible fashion and written in standard English? PLOS ONE does not copyedit accepted manuscripts, so the language in submitted articles must be clear, correct, and unambiguous. Any typographical or grammatical errors should be corrected at revision, so please note any specific errors here.

Reviewer #1: Yes Reviewer #2: Yes

6. Review Comments to the Author. Please use the space provided to explain your answers to the questions above and, if applicable, provide comments about issues authors must address before this protocol can be accepted for publication. You may also include additional comments for the author, including concerns about research or publication ethics.

You may also provide optional suggestions and comments to authors that they might find helpful in planning their study. (Please upload your review as an attachment if it exceeds 20,000 characters)

Response: Please see our response below.

Reviewer #1

Thank you for inviting me to review this study protocol that seeks to an important public health and urban planning question: how micro-scale streetscape characteristics shape walking-related physical activity and how these relationships may differ across socioeconomic groups. The protocol is ambitious and has several notable strengths, including the integration of smartphone-based GPS and step-count data, detailed environmental exposure assessment using street-view imagery and GIS, and a mixed-methods design that combines quantitative analysis with qualitative interviews and co-creation elements. The emphasis on equity is also a clear strength. Overall, this is a strong and interesting protocol with a solid rationale, a well-designed mixed-methods approach, and clear relevance for both research and practice. Below please see my specific concerns.

Response: We sincerely thank you for your thorough review of this manuscript and for your constructive comments.

1. In the data analysis plan section, the authors explain that GPS trajectories will be cleaned and matched with environmental indicators, which is helpful. It would further strengthen the protocol if the authors briefly clarify how exposures will be estimated. A short clarification here would improve transparency and reproducibility.

Response: Done. We have now clarified the exposure assessment approach in the Data Analysis Plan section, specifying how GPS trajectories will be linked to micro-scale environmental indicators across space and time. The following was added: “Accelerometer and GPS data will be matched and integrated with micro-scale built and natural environmental indicators using GIS. For recorded locations, exposures will be assigned based on street segment characteristics, with optional buffering (e.g., 25 m) to account for positional uncertainty. Environmental indicators will then be aggregated to the temporal resolution of the outcome (e.g., half-hour intervals) by computing time-weighted averages along individual mobility paths.”

2. The manuscript already addresses ethics, pseudonymization, GDPR compliance, and secure storage, which is very good. Because GPS trajectories are particularly sensitive, it would be helpful to add one or two more sentences on practical geoprivacy protections, for example how home/work locations or highly identifying path information will be handled in derived datasets and outputs.

Response: Done. We added: “Raw GPS data will not be published, as sensitive locations (e.g., home and work) could be inferred. To protect geoprivacy, data will be geomasked (e.g., by adding random noise) and/or aggregated to coarser spatial units prior to dissemination, preventing re-identification. ”

3. For the qualitative component, a brief note on coder procedures and how qualitative and quantitative findings will be integrated would make the protocol even clearer.

Response: Done. We added the following: “The results will, amongst others, identify participant-defined environmental factors influencing walking practices and experiences. Qualitative findings on the lived experiences of streetscapes will also be mapped against

---

## [Decision Letter · Decision Letter 1]

22 Apr 2026

Healthy living environments create healthy physical activity-related habits (ELEVATE): A protocol for a smartphone-based mixed-method study

PONE-D-26-07970R1

Dear Dr. Helbich,

We’re pleased to inform you that your manuscript has been judged scientifically suitable for publication and will be formally accepted for publication once it meets all outstanding technical requirements.

Kind regards,

Dong Liu, Ph.D.

Academic Editor

PLOS One

Reviewers' comments:

Reviewer's Responses to Questions

Comments to the Author

1. Does the manuscript provide a valid rationale for the proposed study, with clearly identified and justified research questions?

Reviewer #1: Yes

Reviewer #2: Yes

2. Is the protocol technically sound and planned in a manner that will lead to a meaningful outcome and allow testing the stated hypotheses?

Reviewer #1: Yes

Reviewer #2: Yes

3. Is the methodology feasible and described in sufficient detail to allow the work to be replicable?

Reviewer #1: Yes

Reviewer #2: Yes

4. Have the authors described where all data underlying the findings will be made available when the study is complete?

Reviewer #1: Yes

Reviewer #2: Yes

5. Is the manuscript presented in an intelligible fashion and written in standard English?

Reviewer #1: Yes

Reviewer #2: Yes

6. Review Comments to the Author

You may also provide optional suggestions and comments to authors that they might find helpful in planning their study.

Reviewer #1: Thank you for addressing my concerns. I do not have other questions. I think the protocol is ready for publication now.

Reviewer #2: The author has carefully revised the manuscript and fully addressed all the questions and suggestions I put forward in my review, and I recommend acceptance.

7. PLOS authors have the option to publish the peer review history of their article (what does this mean?). If published, this will include your full peer review and any attached files.

Do you want your identity to be public for this peer review? For information about this choice, including consent withdrawal, please see our Privacy Policy.

Reviewer #1: No

Reviewer #2: No

---

## [Editor Report · Acceptance letter]

PONE-D-26-07970R1

PLOS One

Dear Dr. Helbich,

I'm pleased to inform you that your manuscript has been deemed suitable for publication in PLOS One. Congratulations! Your manuscript is now being handed over to our production team.

Kind regards,

on behalf of

Professor Dong Liu

Academic Editor

PLOS One